# Convective updrafts near sea-breeze fronts

Shizuo Fu[1, 2], Richard Rotunno[3], Huiwen Xue[4]

[1]Key Laboratory for Humid Subtropical Eco-Geographical Processes of the Ministry of Education, Fujian Normal University, Fuzhou, China

[2]School of Geographical Sciences, Fujian Normal University, Fuzhou, China

[3]National Center for Atmospheric Research, Boulder, Colorado

[4]Department of Atmospheric and Oceanic Sciences, School of Physics, Peking University, Beijing, China

*Correspondence to*: Shizuo Fu (fusz@fjnu.edu.cn)

**Abstract.** Sea-breeze fronts (SBFs) are frequently found to trigger deep convection. The convective updrafts near the SBF are
critical in this triggering process. Here, the size and strength of the updrafts near an idealized SBF are investigated with large-eddy simulations. A central focus of this study is to compare the updrafts near the SBF, which are substantially affected by the SBF, to the updrafts ahead of the SBF, which develop in a typical convective boundary layer. It is found that the updrafts near the SBF are larger than, but have similar strength to, the updrafts ahead of the SBF. The larger updrafts near the SBF are produced through the merger between the postfrontal streaky structures and the updrafts originating near the SBF. Lagrangian
budget analysis of vertical momentum reveals that the dynamics experienced by the parcels constituting the updrafts near the SBF is almost the same as that ahead of the SBF, so that the strength of the updrafts near the SBF is similar to that ahead of the SBF. It is also found that when the environmental wind is not included, the size and strength of the updrafts near the SBF scale with the boundary-layer height and the convective velocity scale, respectively, like those in the typical convective boundary layer; however, when the environmental wind is included, the aforementioned scaling breaks down. The present
results should also apply to other boundary-layer convergence lines similar to the SBF.

## 1 Introduction

The sea-breeze circulation (SBC) is a local circulation produced by the differential heating between the land and the sea (Miller et al., 2003; Crosman and Horel, 2010). It frequently occurs in coastal regions (Borne et al., 1998; Papanastasiou and Melas, 2009; Perez and Silva Dias, 2017; Shen et al., 2021). SBCs are often found to play important roles in deep-convection initiation
(DCI), leading to heavy precipitation, strong winds and other severe weather (Koch and Clark, 1999; Carbone et al., 2000; Dauhut et al., 2016).

In the presence of a SBC, one can divide the boundary layer into three regions. The first is the sea-breeze front (SBF), which is the leading edge of the sea breeze. The second is the postfrontal region, which is occupied by the sea breeze near the surface and the return flow aloft. The third is the prefrontal region. When a SBC occurs, the land surface is substantially heated, so a
convective boundary layer develops in the prefrontal region. Many studies have found that DCI occurs preferentially near the SBF rather than in the postfrontal or prefrontal regions (Koch and Ray, 1997; Carbone et al., 2000; Dauhut et al., 2016; Park

et al., 2020; Fu et al., 2021). The boundary layer in the postfrontal region is stabilized by the subsidence associated with the return flow (Cuxart et al., 2014), so that the postfrontal region is less favorable for DCI than the SBF.

In a recent study, Fu et al. (2021) found that the updrafts near the SBF are larger and moister than those in the prefrontal region, so that DCI is favored near the SBF rather than in the prefrontal region. They further showed that the updrafts near the SBF are moister because the sea breeze transports moister air from the sea to the SBF. However, they did not explain why the updrafts near the SBF are larger than those ahead of the SBF. In addition, they did not explain why the strengths of the updrafts near the SBF are similar to those ahead of the SBF. In this study, we aim to shed light on both points.

There are observational studies suggesting that the updraft strength near the SBF is similar to that ahead of the SBF. Wood et al. (1999) performed aircraft observations of the SBF. The resolution of their data is as high as 2.5 m, which is sufficiently high to resolve the structure of the SBF as well as the convective updrafts ahead of the SBF. Their results clearly showed that the strengths of the updrafts near the SBF are similar to those ahead of the SBF. Similar results were also shown with aircraft observations at a resolution of 3 m (Kraus et al., 1990; Stephan et al., 1999). We note that previous studies usually focused on the SBF and did not focus on the region ahead of the SBF. As a result, only a limited number of studies analyzed the updrafts near the SBF along with those ahead of the SBF.

The sea breeze is sometimes considered as a density current (Simpson, 1969, 1982), so it is widely assumed that the sea breeze shares the characteristics of the density current. An important characteristic of the density current is that a strong updraft forms near the outflow boundary (Rotunno et al., 1988; Bryan and Rotunno, 2014; Grant and van den Heever, 2016; Fu et al., 2017). It is well known that this strong updraft is produced by the density-current pressure perturbation (see Fig. 2.6 of Markowski and Richardson, 2010). Ahead of the outflow boundary, no such pressure perturbation exists to produce a strong updraft, which means that the updraft near the outflow boundary is much stronger than that ahead of the outflow boundary. Obviously, the prediction of the density-current analogy is not consistent with the aforementioned observational results. Furthermore, the density-current analogy provides no information on the size of the updraft.

Some studies have pointed out that the sea breeze is different from a typical density current. In a modelling study, Robinson et al. (2013) compared the typical density current to the typical sea breeze. For the case of a typical density current, their model simulates a lock-exchange flow with no surface heating. Their results showed that the near-surface temperature is nearly constant behind the outflow boundary and displays a distinct jump across the outflow boundary. In this situation, the outflow boundary propagates at the speed expected for a typical density current, e.g., that predicted by Benjamin (1968). For the case of a typical sea breeze, the model starts with a horizontally homogeneous profile and has continuous surface heating over the land. Their results showed that the near-surface temperature continuously increases from the coast to the SBF, and the temperature difference across the SBF is very small. In this situation, the SBF propagates at a speed less than that expected for a typical density current. They concluded that the continuous surface heating causes the sea breeze to behave differently from a typical density current. This conclusion is also supported by observational studies (Reible et al., 1993; Carbone et al., 2000). Based on the discussions above, it appears that the characteristics of the convective updrafts near the SBF are not well understood. In this study, we seek to improve that understanding. Since we want to understand why the SBF is the more

favorable region for DCI than the prefrontal region, we compare the characteristics of the updrafts near the SBF to those ahead of the SBF. We also compare the Lagrangian dynamics of parcels that constitute the updrafts near the SBF to that of updrafts ahead of the SBF. In Sect. 2, we present our analysis methods while Sect. 3 discusses the results. The conclusions are presented in Sect. 4.

## 2 Methods

### 2.1 Model and experimental setup

The present simulations were performed with the release 19.10 of Cloud Model 1 (CM1; Bryan and Fritsch, 2002). The compressible governing equations are solved with the time-splitting algorithm (Klemp and Wilhelmson, 1978), in which acoustic waves are solved explicitly in the horizontal direction, and implicitly in the vertical direction. Subgrid-scale turbulence is represented by the turbulence kinetic energy (TKE) scheme (Deardorff, 1980). In this study, we focus on the processes taking place before DCI, so we do not consider moist processes. Please refer to Fu et al. (2021) for results with moist processes. In addition, we do not consider the Coriolis force or radiative transfer.

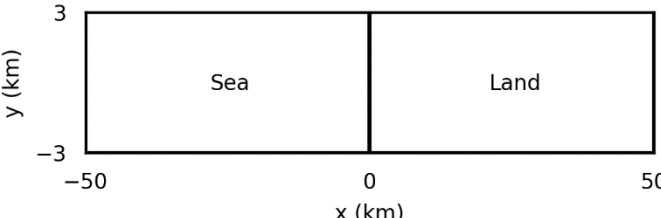

**Figure 1: Domain configuration.**

Figure 1 shows the domain configuration. The sea and land are located on the left and right halves of the domain, respectively. In the cross-coast ($x$) direction, the domain size is 100 km. The resolution in the $x$-direction is constant at 20 m over the land; over the sea, the resolution is 20 m at the coast, and then gradually stretches to 180 m at the left boundary. "Open" boundary conditions are used in the $x$-direction. In addition, Rayleigh damping on all fields is applied at $x < -45$ km and $x > 45$ km. In the along-coast ($y$) direction, the domain size is 6 km; the resolution is constant at 20 m and periodic boundary conditions are used. In the vertical ($z$) direction, the domain size is 3.4 km; the resolution is 20 m below $z = 1.4$ km, then gradually stretches to 60 m at $z = 2.2$ km, and remains at 60 m up to the model top. Rayleigh damping is applied above $z = 2.5$ km.

The initial profile follows that of Sullivan and Patton (2011). Below $z = 974$ m, the potential temperature is constant at 300 K, indicating a neutral layer. From $z = 974$ m to 1074 m, the potential temperature increases by 0.08 K m$^{-1}$. This is a very stable layer, which strongly limits the deepening of the convective boundary layer. Above $z = 1074$ m, the potential temperature increases by 0.003 K m$^{-1}$. This initial profile is representative of atmospheric profiles several hours after sunrise on a clear-sky day.

The land and the sea are distinguished by sensible heat flux (SHF) and roughness length. Over the sea, the SHF is usually very small (Yu and Weller, 2007). Thus, a zero SHF is prescribed, as is done in previous idealized simulations of SBCs (Antonelli and Rotunno, 2007; Crosman and Horel, 2010, 2012). Over the land, three values are considered for SHF, i.e., 0.1, 0.2, and 0.3 K m s$^{-1}$. Hereafter, the three simulations are referred to as SHF01, SHF02, and SHF03, respectively. Note that SHF is strongly affected by solar radiation, which varies substantially with cloud fraction and/or season. As a result, SHF ranging from 0.06 to 0.3 K m s$^{-1}$ can be found in the literature (Crosman and Horel, 2012). The roughness length is set to $2 \times 10^{-4}$ m over the sea and 0.1 m over the land (Wieringa, 1993).

The environmental wind is another factor that substantially affects the SBC (Bechtold et al., 1991; Miller et al., 2003; Crosman and Horel, 2010). When the environmental wind is strong, no SBC develops. When the environmental wind is weak, an offshore wind shifts the SBF seaward and enhances the SBC; while an onshore wind shifts the SBF inland and weakens the SBC. Due to the constraints of computational resource, we perform only two sensitivity tests regarding environmental wind. The first has an offshore (negative) environmental wind of 2 m s$^{-1}$, and the second has an onshore (positive) environmental wind of 2 m s$^{-1}$. The SHF is 0.2 K m s$^{-1}$ in both simulations. These two simulations are referred to as SHF02_Un2 (n2 means negative 2) and SHF02_Up2 (p2 means positive 2), respectively.

Online Lagrangian parcels are used to investigate the development of the updrafts. The resolved velocity is tri-linearly interpolated to the positions of the parcels, and then used to update the positions of the parcels at each time step. The subgrid-scale velocity is not included in this calculation. Yang et al. (2008) pointed out that the single-particle dispersion can be accurately modelled by LES because the errors in the Lagrangian velocity correlation and the Lagrangian velocity fluctuation tend to cancel each other. In addition, the model resolution in this study is so high that the subgrid-scale TKE is much smaller than the resolved TKE. In this situation, the effect of subgrid-scale velocity on parcel trajectory should be weak (Yang et al., 2015).

Each simulation is run for 4 h. The three-dimensional (3-D) fields were saved every 10 min. Similar to Fu et al. (2021), it was found that the 10-min data was not sufficient to resolve the fast evolution of the updrafts. Therefore, the model was restarted and the 3-D fields were saved every 1 min. In addition, the Lagrangian parcels quickly form clusters after being released, so their spatial representativeness declines. In order to mitigate this effect, we reset the positions of the parcels when we restart the model. At each restart, parcels are released in each grid cell in the region of $x_{SBF} - 5 \text{ km} < x < x_{SBF} + 15 \text{ km}$, -3 km $< y <$ 3 km, and 0 km $< z <$ 1 km, where $x_{SBF}$ is the position of the SBF. This region is large enough so that the updrafts being investigated are always densely populated with parcels throughout the tracking. The model was restarted at four times, i.e., $t$ = 2 h, 2 h 30 min, 3 h, and 3 h 30 min.

## 2.2 Procedures for compositing updrafts

Due to the turbulent nature of the flow, it is difficult, and probably not useful, to analyze the characteristics of individual updrafts. Therefore, the updrafts near and ahead of the SBF are separately composited, and the characteristics of the composite

updrafts are analyzed. The compositing procedure is generally similar to that used by Fu et al. (2021), but with some
modifications.

The position of the SBF is first defined. Since the simulation setup is homogeneous in the $y$-direction, we define the position of the SBF only in the $x$-direction. The cross-coast wind at $z = 0.21$ km is averaged over the $y$-direction. A running average is performed twice to filter out the turbulence. The window for the running average is 2 km. The filtered cross-coast wind is used to calculate the horizontal convergence. The position having the maximum horizontal convergence is defined as the position
of the SBF.

The updrafts are then defined. In the horizontal cross section at $z = 0.5z_i$, any grid point with vertical velocity greater than $0.8w^*$ is defined as within an updraft. The boundary-layer height $z_i$ is defined as the height of the lowest grid point with $d\bar{\theta}/dz$ > 3 K km$^{-1}$. The mean potential temperature $\bar{\theta}$ is calculated in the region from $x_{SBF,E} + 2$ km to $x_{SBF,E} + 7$ km, where $x_{SBF,E}$ is the position of the SBF at the end of the simulation. Note that this region is not affected by the SBC throughout the simulations.
The convective velocity scale $w^*$ is defined as

$$w^* = \left(\frac{g}{\theta_0}\overline{w'\theta'}z_i\right)^{1/3}, (1)$$

where $g$ is the gravitational acceleration, $\theta_0$ the reference potential temperature, and $\overline{w'\theta'}$ the SHF. All grid points that are identified as within an updraft are four-way connected to form clusters. Each cluster defines an updraft. When defining the updrafts, we rely on two parameters, i.e., the height $z = 0.5z_i$, where the updrafts are defined, and the threshold vertical velocity
$0.8w^*$, above which a grid point is defined as within an updraft. Sensitivity tests show that the results are qualitatively the same when the height is changed to $0.3z_i$ or $0.7z_i$, and when the threshold vertical velocity is changed to $1.0w^*$ or $1.2w^*$.

We then define the position of an updraft as the centroid of the updraft, which is the mean horizontal position of all grid points within the updraft at $z = 0.5z_i$. Furthermore, an updraft is defined as a frontal updraft if it is less than 1 km away from the SBF; an updraft is defined as a prefrontal updraft if it is more than 1 km ahead of the SBF. In order to accelerate the calculation,
only those prefrontal updrafts whose positions are between $x_{SBF,E} + 2$ km and $x_{SBF,E} + 7$ km are considered in the compositing procedure.

It is found that some updrafts are very small, and should not trigger a convective cell. Thus, updrafts with areas smaller than $4 \times 10^4$ m$^{-2}$ are excluded. It is also found that some updrafts are very close to each other, and should not be considered as independent updrafts. Therefore, the distance between any possible pair of updrafts is calculated. If the distance between a pair
of updrafts is less than the boundary-layer height, the smaller updraft is excluded.

Finally, the method introduced by Schmidt and Schumann (1989) is used to composite the updrafts. In order to composite the frontal updrafts, they are shifted horizontally so that their centroids coincide. Ensemble averaging is then conducted over all these coincided frontal updrafts to produce the composite frontal updraft. Note that each centroid indicates a 3-D updraft, so the foregoing procedure produces a 3-D composite frontal updraft. All the frontal updrafts identified from the 1-min data
between $t = 2$ and 4 h are included in the compositing procedure. The prefrontal updrafts are composited similarly.

## 3 Results

The results of the five simulations are qualitatively similar. In this section, we first detail the results of simulation SHF02, and then discuss the results of the other four simulations.

### 3.1 Structure of the SBC

Figure 2 shows the along-coast averaged potential temperature, full pressure perturbation, and cross-coast wind at $t = 2$ h in simulation SHF02. The position of the SBF is shown with the dashed lines. At this time, the sensible heating increases the temperature ahead of the SBF by approximately 1.5 K; while the temperature over the sea remains almost the same as the initial condition. In addition, the temperature increases smoothly from the coast ($x = 0$) to the SBF, similar to that found in previous studies (Reible et al., 1993; Robinson et al., 2013). The temperature difference accounts for the pressure perturbation

(Fig. 2b). No pressure perturbation extremum is found near the SBF, consistent with the finding of Robinson et al. (2013). Figure 2c shows the sea breeze near the surface and the return flow aloft, which implies a deep shear layer behind the SBF. At $t = 2$ h, the SBF is at $x = 7.8$ km; and the boundary-layer height is 0.99 km in the prefrontal region. In the following 2 h, the SBF continuously moves inland and the boundary layer slowly deepens. At $t = 4$ h, the SBF reaches $x = 20.1$ km; and the boundary-layer height reaches 1.15 km.

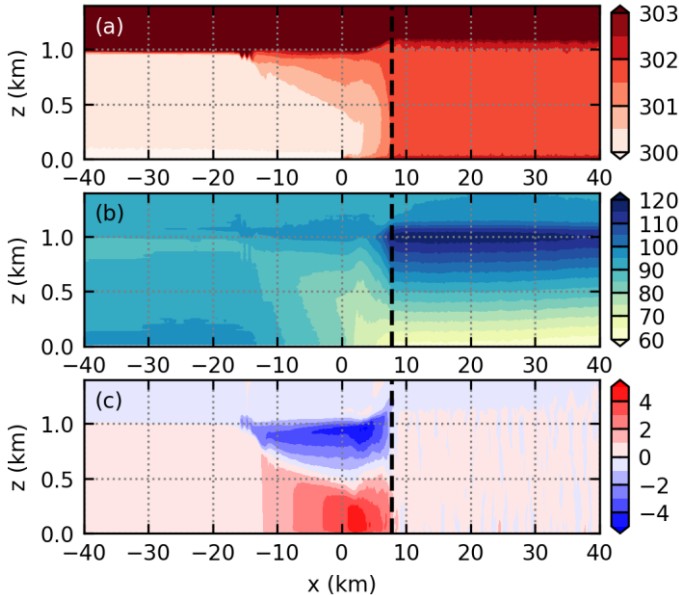


**Figure 2: Along-coast averaged (a) potential temperature (K), (b) full pressure perturbation (Pa), and (c) cross-coast wind (m s⁻¹) at $t = 2$ h in simulation SHF02. The dashed lines indicate the position of the SBF.**

**3.2 Formation of large updrafts near the SBF**

Figure 3 shows a horizontal cross section of the cross-coast wind and the vertical velocity at $z = 0.5z_i$ and at $t = 2$ h 48 min in
simulation SHF02. Figure 3a shows that streaky structures of positive cross-coast wind are produced behind the SBF. Previous
studies have proposed several theories explaining the formation of streaky structures, e.g., the inflection-point instability and
the convective instability (Etling and Brown, 1993; Gryschka and Raasch, 2005). The inflection-point instability relies on a
sufficiently strong inflection point in the along-coast wind profile. However, no such inflection point exists in our simulations
(not shown). The convective instability is usually measured with a parameter $-z_i/L$, where $L$ is the Obukhov length (Khanna
and Brasseur, 1998; Salesky et al., 2017). In the region from $x = 10$ to 12 km and from $y = -3$ to 3 km, calculation shows that
the mean value of $-z_i/L$ is 70. Based on previous studies, this value should correspond to cells, instead of the streaky
structures shown in Fig. 3a. Previous studies suggest that the threshold values of $-z_i/L$ hold for situations where the shear is
limited near the surface; while in our simulations, the shear occurs over a deep layer (Fig. 2c). It is interesting to mention that
some studies suggest that wind shear alone is sufficient for the generation of streaky structures (e.g., Lee et al., 1990).

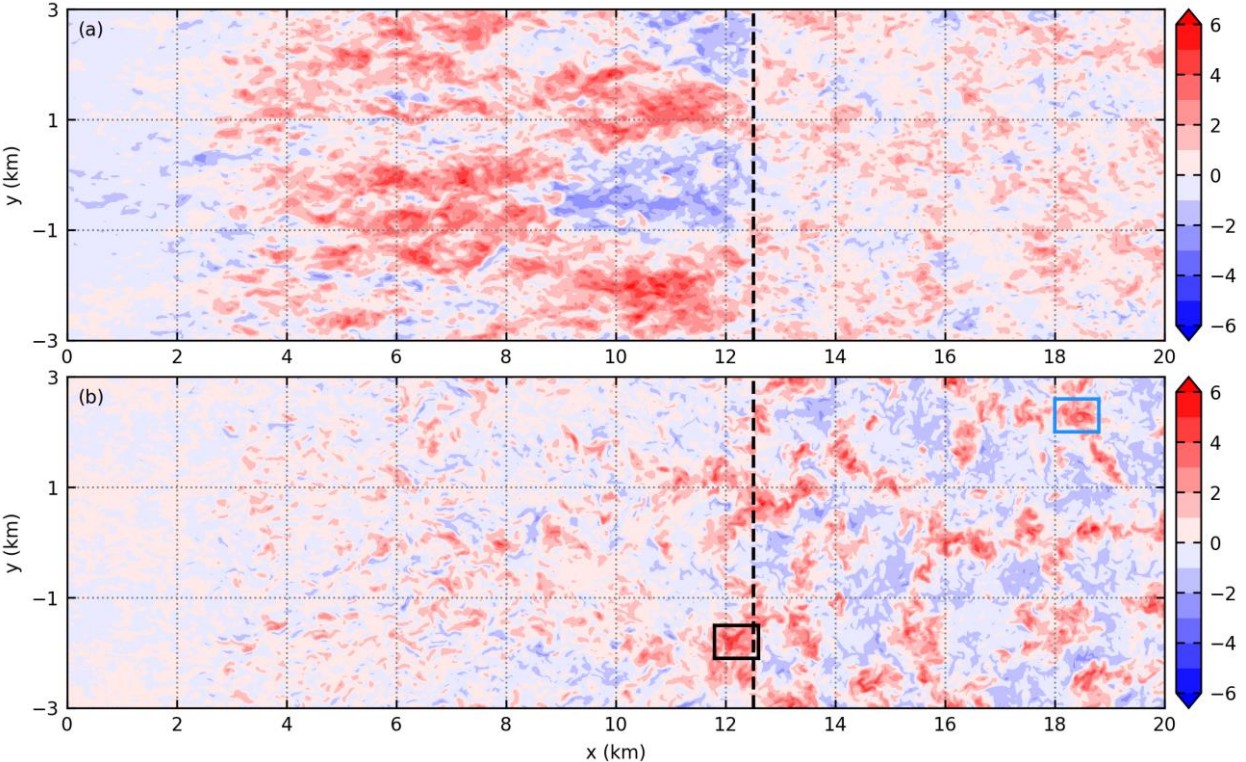


**Figure 3: A horizontal cross section of (a) cross-coast wind (m s⁻¹) and (b) vertical velocity (m s⁻¹) at $z = 0.5z_i$ and at $t = 2$ h 48 min in simulation SHF02. The dashed lines indicate the position of the SBF. The rectangles enclose the updrafts whose formation processes are investigated with parcel trajectories.**

Figure 3b indicates that the updrafts near the SBF are larger than those ahead of the SBF. A comparison of Figs. 3a and 3b reveals that the larger updrafts near the SBF are closely related to the postfrontal streaky structures. Note that Fig. 3 is a snapshot representative of all times when large updrafts are visible (see supplement). Figure 3b also shows that the updrafts far behind the SBF are generally much weaker than those near or ahead of the SBF. They are less likely to trigger convective cells and are hence not analyzed in detail.

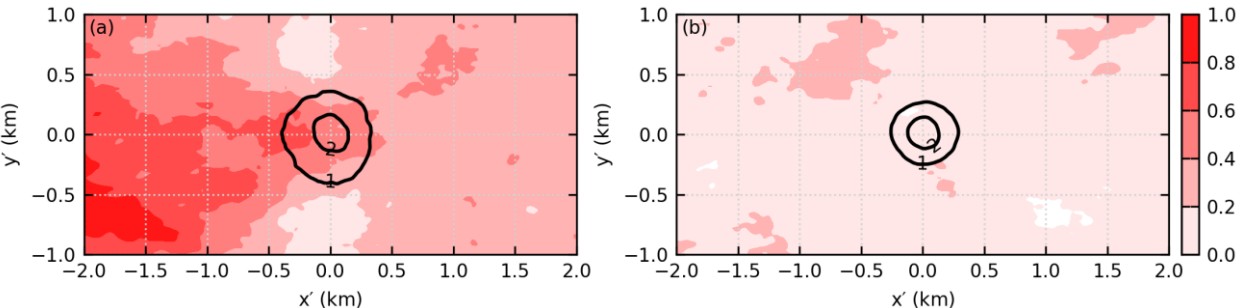

Figure 4: (a) Horizontal cross section of vertical velocity (m s$^{-1}$; black contour) and cross-coast wind (m s$^{-1}$; filled contour) of the composite frontal updraft at $z = 0.5z_i$ in simulation SHF02. (b) The same as (a), except for the composite prefrontal updraft.

Figure 4a shows the vertical velocity and the cross-coast wind of the composite frontal updraft at $z = 0.5z_i$. It clearly shows that the frontal updraft forms at the leading edge of the streaky structure, confirming the snapshot impression from Fig. 3. As a comparison, Fig. 4b shows that no similar structure exists near the prefrontal updraft.

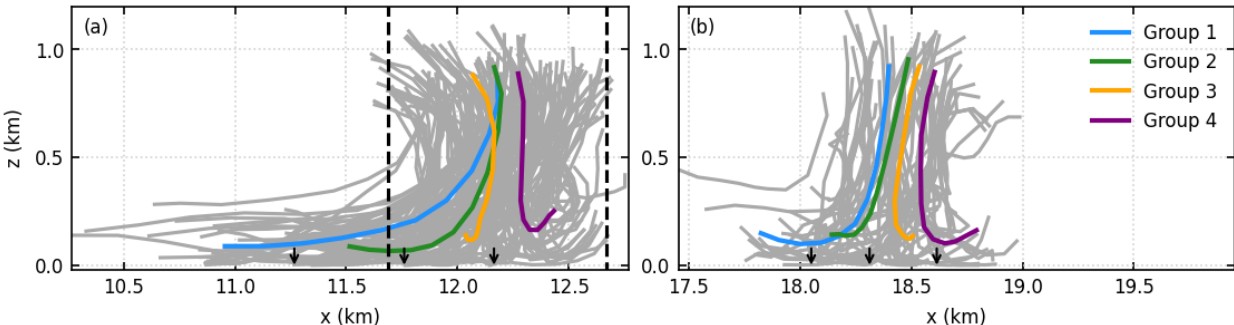

Figure 5: (a) Gray lines are individual parcel trajectories that cross the region enclosed by the black rectangle in Fig. 3b. The parcels are tracked from $t = 2$ h 40 min to 2 h 50 min in simulation SHF02. Every 50th parcel trajectory is shown. The four colored lines show the mean parcel trajectories of the four groups, respectively. The three arrows show the $x$-positions that separate the trajectories into four groups. The left and right dashed lines indicate the position of the SBF at $t = 2$ h 40 min and 2 h 50 min, respectively. (b) The same as (a), except for parcel trajectories that cross the region enclosed by the blue rectangle in Fig. 3b.

Parcel trajectories are used to further analyze how the updrafts are produced. A parcel is said to cross the height $z$ at time $t$, if it is below the height $z$ at time $t - 1$ min, above $z$ at time $t$, and ascends by more than 0.12 km from time $t - 1$ min to $t$ (corresponding to a vertical velocity of 2 m s$^{-1}$). The gray lines in Figs. 5a and 5b show the parcel trajectories that cross the height $z = 0.5z_i$ at $t = 2$ h 48 min through the region enclosed by the black rectangle and blue rectangle in Fig. 3b, respectively. The parcels are tracked from $t = 2$ h 40 min to 2 h 50 min. In order to know where the parcels come from, e.g., from behind

the SBF or from ahead of the SBF, they are divided into four groups by the 25th, 50th, and 75th percentiles of their $x$-positions at $t$ = 2 h 40 min. The demarcations separating the four groups are shown with the black arrows. The colored lines in Fig. 5 show the mean parcel trajectories of each group.

Ahead of the SBF (Fig. 5b), the parcels mostly ascend vertically, except near the surface, where the horizontal convergence of 215 parcels is apparent and well understood (Stull, 1988). Near the SBF (Fig. 5a), there also exist parcels that ascend vertically, e.g., the parcels in the third and fourth groups. In addition to this, many parcels ascend along slanted trajectories, e.g., the parcels in the first and second groups. These parcels gain buoyancy from behind the SBF. They are then transported toward the SBF by the sea breeze, and merge with the parcels that originate near the SBF. Note that the parcels rising from behind the SBF are part of the streaky structure. This suggests that the larger updraft near the SBF forms as a result of the merger between 220 the streaky structure and the updraft that originates near the SBF.

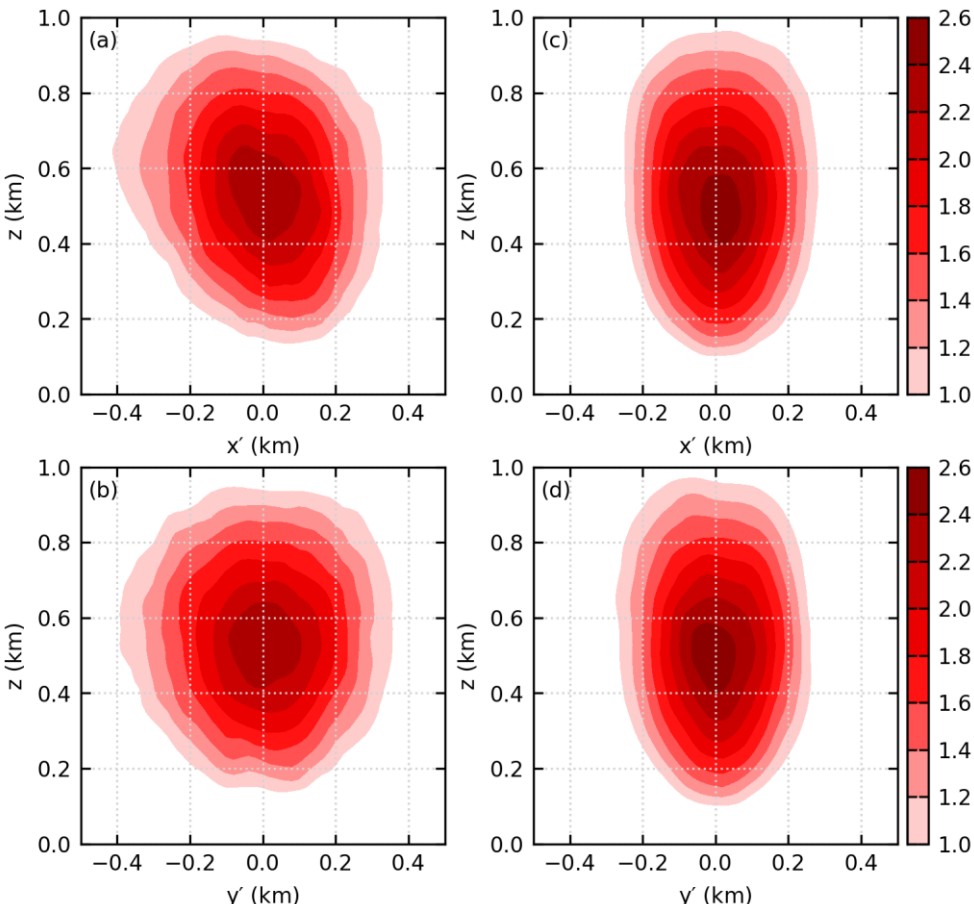

Figure 6: (a) $x$-$z$ cross section and (b) $y$-$z$ cross section of the vertical velocity (m s$^{-1}$) of the composite frontal updraft in simulation SHF02. (c) and (d) are the same as (a) and (b), except for the composite prefrontal updraft.

The updrafts near the SBF and those ahead of the SBF are separately composited and shown in Fig. 6. It shows that the
composite frontal updraft is larger than the composite prefrontal updraft, both in the $x$-$z$ plane (cf. Figs. 6a and 6c) and in the
$y$-$z$ plane (cf. Figs. 6b and 6d), consistent with the finding of Fu et al. (2021). It is also found that the maximum vertical
velocity of the composite frontal updraft is similar to that of the composite prefrontal updraft, with a difference less than 10%,
again consistent with the finding of Fu et al. (2021). We note that the resolution used by Fu et al. (2021) is 100 m in the
horizontal and 40 m in the vertical, while the resolution used in this study is 20 m both in the horizontal and in the vertical.
This means that our conclusion is independent of model resolution.

### 3.3 Lagrangian budget analysis of vertical momentum

The budget of vertical momentum along the parcel trajectories is analyzed to investigate whether the dynamical forcing of the
updrafts near the SBF is different from that ahead of the SBF. The Lagrangian vertical momentum equation is

$\frac{dw}{dt} = b_{eff} - \frac{1}{\bar{\rho}} \frac{\partial p'_d}{\partial z}$, (2)

where $w$ is vertical velocity, $\bar{\rho}$ the reference density, and $p'_d$ the dynamic pressure perturbation. The effective buoyancy $b_{eff}$
is

$b_{eff} = b - \frac{1}{\bar{\rho}} \frac{\partial p'_b}{\partial z}$, (3)

where $b$ is buoyancy, and $p'_b$ the buoyancy pressure perturbation. The pressure perturbations satisfy

$\nabla^2 p'_d = -\nabla \cdot (\bar{\rho} \boldsymbol{v} \cdot \nabla \boldsymbol{v})$, and (4)

$\nabla^2 p'_b = \frac{\partial \bar{\rho} b}{\partial z}$, (5)

where $\boldsymbol{v}$ is the velocity vector. Following Markowski and Richardson (2010, p.29), $p'_b$ is calculated by solving Eq. (5), and $p'_d$
is then obtained by subtracting $p'_b$ from the full pressure perturbation.

In the Lagrangian budget analysis, we include only those parcels that continuously ascend to the top of the boundary layer.
For each parcel, the first time it rises above $z = 0.9$ km is defined as $t_{top}$. We then search backward in time to find the period
during which the parcel ascends continuously and define the start of this period as $t_{lift}$. The history between $t_{lift}$ and $t_{top}$ is
used for the budget analysis. In addition, a parcel is defined as near the SBF if its $x$-position ($x_p$) satisfies $x_{SBF} - 1$ km $< x_p <$
$x_{SBF} + 1$ km throughout the continuously ascending period. A parcel is defined as ahead of the SBF if it satisfies $x_{SBF,e} + 5$
km $< x_p < x_{SBF,e} + 10$ km throughout the ascending period. $x_{SBF,e}$ is the position of the SBF at 30 min after the release of the
parcels.
Figure 7 shows the profiles of effective buoyancy and dynamic pressure gradient force for parcels released at $t = 2$ h and
tracked for 10 min in simulation SHF02. The effective buoyancy is positive from the surface up to $z = 0.9$ km, and then
becomes negative; and dynamic pressure gradient force is positive from the surface up to $z = 0.6$ km, and then becomes
negative. The profiles of both the effective buoyancy and the dynamic pressure gradient force are similar to those of Torri et
al. (2015). More importantly, the dynamics experienced by the parcels near the SBF is almost the same as that experienced by

the parcels ahead of the SBF. We note that the results are similar for parcels that are released at $t = 2$ h and tracked for 15 min (not shown); and the results are also similar for parcels that are released at $t = 2$ h 30 min, 3 h, and 3 h 30 min, either tracked for 10 or 15 min (not shown). The similar dynamics explains the fact that the strengths of the updrafts near the SBF are similar to those ahead of the SBF. Figure 7 also shows that there is no extra dynamic pressure gradient force near the SBF. This is also different from the density-current analogy.

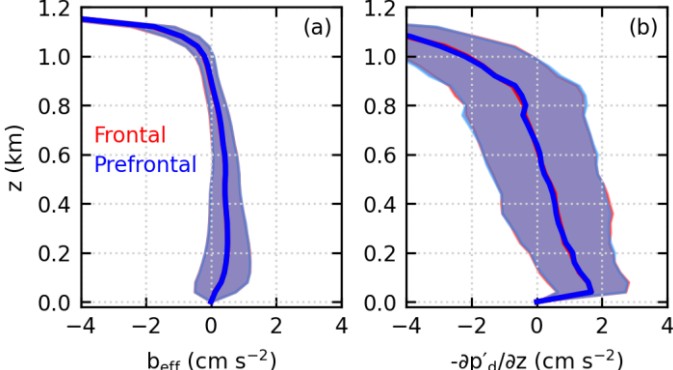


**Figure 7: Profiles of (a) effective buoyancy and (b) dynamic pressure gradient force along the parcel trajectories. The parcels are released at $t = 2$ h and tracked for 10 min in simulation SHF02. The solid lines show the averages, and the shadings show the standard deviations. Note that the profiles near the SBF are almost the same as those ahead of the SBF.**

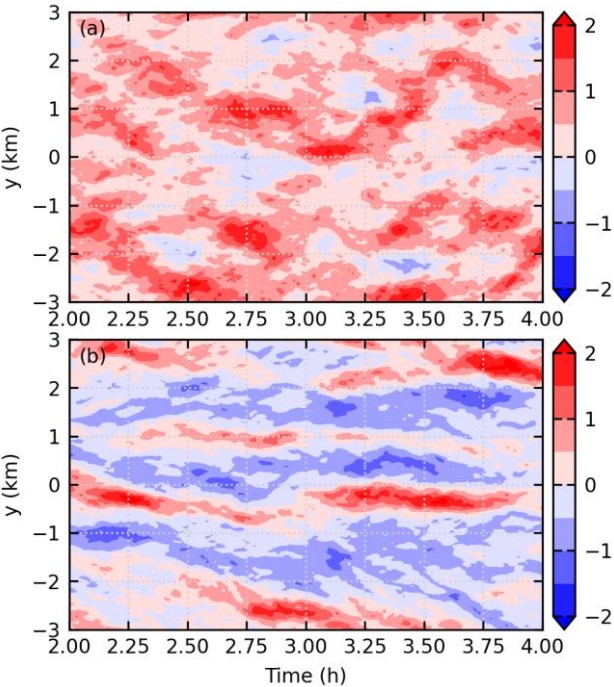

**Figure 8: Temporal evolution of vertical velocity (m s⁻¹) at $z = 0.5z_i$ averaged (a) from $x_{SBF}$ - 1 km to $x_{SBF}$ + 1 km and (b) from $x =$ 30 to 32 km in simulation SHF02.**

## 3.4 Persistence of the updrafts

The persistence of an updraft also affects its potential in DCI. Figure 8a shows the temporal evolution of the vertical velocity at $z = 0.5z_i$ averaged from $x_{SBF} - 1$ km to $x_{SBF} + 1$ km and Fig. 8b shows that averaged from $x = 30$ to 32 km, a region that is ahead of the SBF throughout the simulation. It is seen that the updrafts near the SBF are generally shorter-lived than those ahead of the SBF, suggesting that the persistence of updrafts cannot explain the fact that the SBF is the more favorable region for DCI than the prefrontal region. Nevertheless, it is worth pointing out that the lifetime of the updrafts near the SBF is long enough for parcels near the surface to be lifted to the top of the boundary layer, as can be seen from Fig. 5.

## 3.5 Sensitivity to SHF and environmental wind

We now discuss the sensitivity of the results to SHF and environmental wind. Table 1 lists the $x_{SBF}$, $z_i$, and $w^*$ at the end of the simulations. It also lists the horizontal convergence at the SBF averaged from $t = 2$ to 4 h. The horizontal convergence is calculated as described in Sect. 2.2. In the simulations without environmental wind (i.e., simulations SHF01, SHF02, and SHF03), increasing SHF from 0.1 to 0.3 K m$^{-1}$ increases the propagation speed of the SBF (Antonelli and Rotunno, 2007), so the SBF moves farther inland by the end of the simulations; increasing SHF also increases the boundary-layer height, the convective velocity scale, and the mean horizontal convergence at the SBF.

Compared to simulation SHF02, the SBF propagates at a slower speed in simulation SHF02_Un2, but propagates at a faster speed in simulation SHF02_Up2, consistent with previous studies (Miller et al. 2003). Thus, the SBF moves less inland in simulation SHF02_Un2 while farther inland in simulation SHF02_Up2 than that in simulation SHF02. Table 1 also indicates that neither the boundary-layer height nor the convective velocity scale is sensitive to the environmental wind. However, the mean horizontal convergence at the SBF is sensitive to the environmental wind: a negative environmental wind increases the convergence while a positive environmental wind decreases the convergence.

In all simulations considered in this study, the frontal updrafts are produced at the leading edge of the postfrontal streaky structures (not shown), and the dynamics experienced by the parcels constituting the updrafts near the SBF is nearly the same as that ahead of the SBF (not shown), as seen in simulation SHF02.

Table 1. The position of the SBF ($x_{SBF}$), boundary-layer height ($z_i$), and convective velocity scale ($w^*$) at the end of the simulations, and the mean horizontal convergence at the SBF averaged from $t = 2$ to 4 h.

| Simulation | $x_{SBF}$ (km) | $z_i$ (km) | $w^*$ (m s$^{-1}$) | Convergence ($10^{-3}$ s$^{-1}$) |
|---|---|---|---|---|
| SHF01 | 14.7 | 1.03 | 1.50 | 0.85 |
| SHF02 | 20.1 | 1.15 | 1.96 | 1.04 |
| SHF03 | 24.4 | 1.31 | 2.34 | 1.14 |
| SHF02_Un2 | 13.8 | 1.13 | 1.95 | 1.52 |
| SHF02_Up2 | 32.7 | 1.15 | 1.96 | 0.53 |

In a classical convective boundary layer, such as that ahead of the SBF, it is well-known that the size of the updraft scales with $z_i$ and the strength of the updraft scales with $w^*$ (Stull, 1988). Since the dynamics of the frontal updrafts is similar to that of the prefrontal updrafts, it is expected that this scaling is also applicable for frontal updrafts. In order to test this speculation, we re-composited the updrafts using a slightly different procedure. At each output time, the size is normalized by $z_i$ and the vertical velocity is normalized by $w^*$ before the ensemble averaging. The other steps are the same as those described in Sect. 2.2.

We first consider the three simulations without environmental wind, which are shown with Fig. 9. Ahead of the SBF (Figs. 9c and 9d), the composite normalized updrafts are similar for all three simulations, as is well-known. More importantly, the composite normalized updrafts near the SBF are also similar for all three simulations (Figs. 9a and 9b). This means that the aforementioned scaling also works for the frontal updrafts when the environmental wind is zero. Note that although the composite normalized updrafts are similar in both size and strength in all three simulations, the composite dimensional updrafts actually become larger and stronger as SHF increases, as can be deduced from the increasing $z_i$ and $w^*$ (Table 1). In each simulation, Fig. 9 also shows that the frontal updraft is larger than the prefrontal updraft and their strengths are similar.

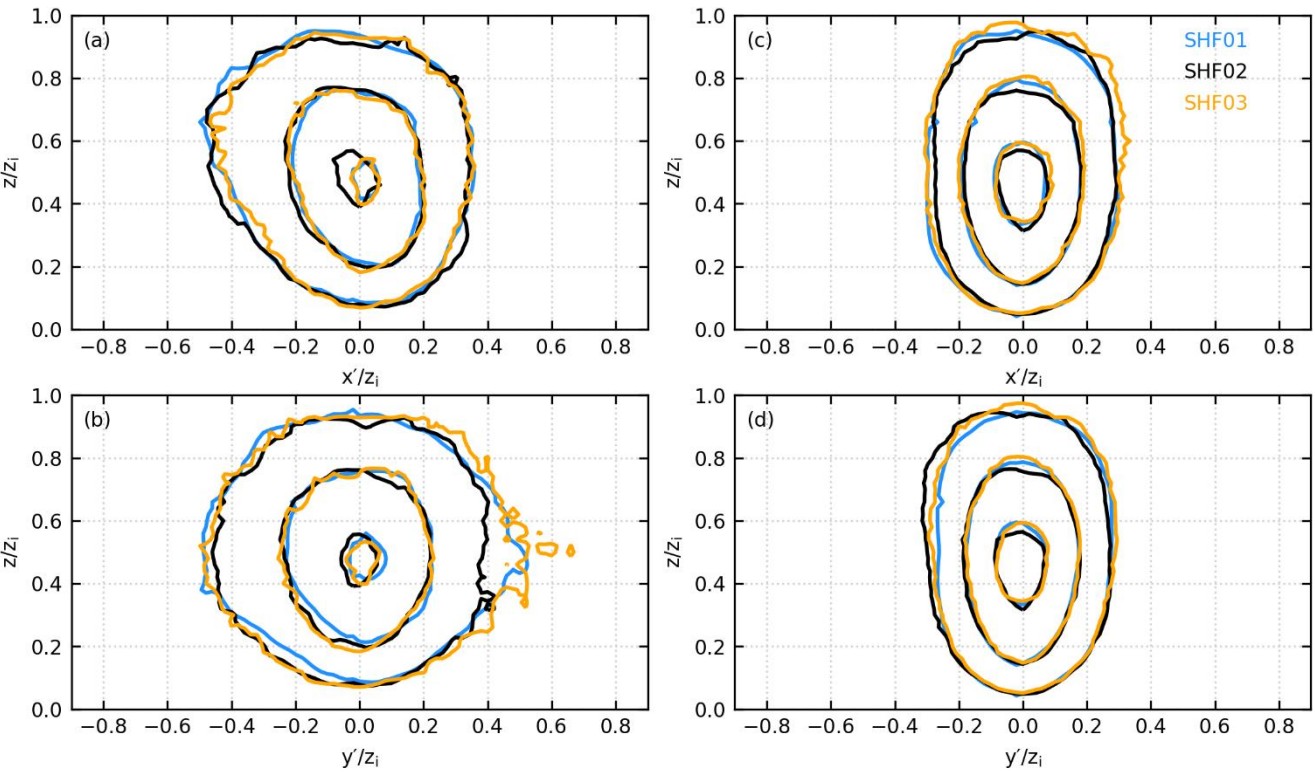

**Figure 9: (a) $x$-$z$ cross section and (b) $y$-$z$ cross section of the normalized vertical velocity of the composite normalized frontal updraft in simulations SHF01, SHF02, and SHF03. The contour levels are 0.4, 0.8, and 1.2. (c) and (d) are the same as (a) and (b), except for the composite normalized prefrontal updraft.**

We next consider the three simulations with varying environmental wind, which are shown with Fig. 10. Ahead of the SBF, the composite normalized updrafts are similar in the $y$-$z$ cross section for all three simulations (Fig. 10d). In the $x$-$z$ cross section (Fig. 10c), the updraft slightly tilts downwind due to the environmental wind. Near the SBF, Figs. 10a and 10b show that the composite normalized frontal updrafts are substantially different among the three simulations. In the $x$-$z$ cross section (Fig. 10a), the frontal updrafts tilt to the left in all three simulations; however, the frontal updraft is less tilted in simulation

SHF02_Up2 and is more tilted in simulation SHF02_Un2. This is because a positive environmental wind reduces the wind shear while a negative environmental wind enhances the wind shear near the SBF.

In the $y$-$z$ cross section (Fig. 10b), the composite normalized frontal updrafts are similar in the $z$-direction, but are very different in the $y$-direction. In simulation SHF02_Un2, the negative wind enhances the convergence near the SBF (Table 1). More updrafts are therefore transported toward the SBF. Their merger produces a wider updraft in the $y$-direction. In

simulation SHF02_Up2, the positive environmental wind weakens the convergence near the SBF (Table 1). Fewer updrafts are transported to the SBF, and their merger produces a narrower updraft in the $y$-direction. Nevertheless, Fig. 10 clearly shows that the frontal updrafts are larger than, and have similar strength to, the prefrontal updrafts in all three simulations with environmental wind.

Based on the results of Antonelli and Rotunno (2007), the $y$-averaged cross-coast wind $\bar{u} \sim \left( \frac{g}{\theta_0} \overline{w'\theta'} \right)^{1/2} t^{1/2} (Nt)^{0.1}$ and

$z_i \sim \left( \frac{g}{\theta_0} \overline{w'\theta'} \right)^{1/2} t^{3/2} (Nt)^{-1}$, where $N$ is the Brunt-Väisälä frequency. In this study, we do not vary $N$ and we compare the simulation results at the same time $t$, so we drop the nondimensional factor $Nt$ from the aforementioned scaling, leading to

$\bar{u} \sim \left( \frac{g}{\theta_0} \overline{w'\theta'} \right)^{1/2} t^{1/2}$, and (6)

$z_i \sim \left( \frac{g}{\theta_0} \overline{w'\theta'} \right)^{1/2} t^{3/2}$. (7)

Substituting Eq. (7) into Eq. (1), we obtain $\bar{u} \sim w^*$.

Based on our analysis in Sect. 3.2, it is the convergence that affects the size of the frontal updrafts. The convergence near the SBF is measured by $\frac{\partial \bar{u}}{\partial x}$. When the environmental wind is zero, the convergence $\frac{\partial \bar{u}}{\partial x} \sim \frac{\bar{u}}{W_{SBF}}$, where $W_{SBF}$ is the width of the SBF. Since $\bar{u} \sim w^*$ and Fig. 9 suggests that the effect of convergence does not introduce scales other than $z_i$ and $w^*$, we obtain that $W_{SBF} \sim z_i$. This means that the width of the SBF is approximately 1 km in this study (Table 1). The observational study by Chiba (1993) found that the width of the SBF was between 0.13 and 1.12 km. When the environmental wind is not zero, the

convergence $\frac{\partial \bar{u}}{\partial x} \sim \frac{\bar{u} - U}{W_{SBF}}$, where $U$ is the environmental wind. Since $U$ is an independent parameter, it does not scale with $w^*$. As a result, $\bar{u} - U$ does not scale with $w^*$, either. In this situation, the simple scaling breaks down, as seen in Fig. 10.

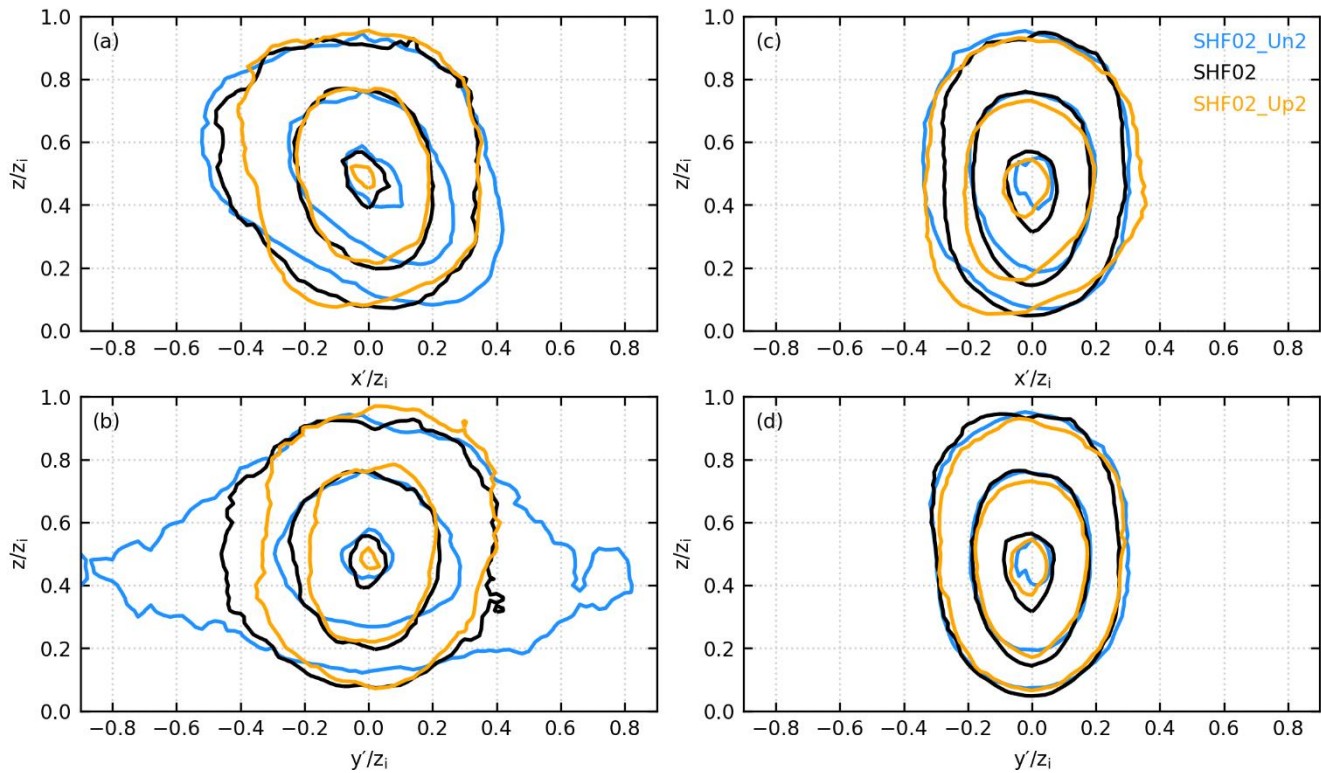

**Figure 10: The same as Fig. 9, except for simulations SHF02_Un2, SHF02, and SHF02_Up2.**

## 4 Conclusions

The sea-breeze circulation (SBC) is frequently found to play an important role in deep-convection initiation (DCI). Previous studies have found that the sea-breeze front (SBF) is a more favorable region for DCI than the prefrontal region. A recent study by Fu et al. (2021) showed that the updrafts near the SBF are larger and moister than the updrafts ahead of the SBF, so that DCI occurs preferentially near the SBF. However, they did not explain why the updrafts near the SBF are larger, and they did not explain why the updrafts near and ahead of the SBF have similar strengths.

This study performs a series of large-eddy simulations to investigate the size and strength of the updrafts near the SBF, and to compare the characteristics of updrafts near the SBF to those ahead of the SBF. Similar to Fu et al. (2021), it is found that the updrafts near the SBF are larger than those ahead of the SBF. It is further shown here that the larger updrafts near the SBF are produced through the merger between the postfrontal streaky structures and the updrafts that originate near the SBF. It is also shown that the updrafts near the SBF have similar strengths to those ahead of the SBF, consistent with the finding of Fu et al.

(2021). This is further investigated here through a Lagrangian budget analysis of the vertical-momentum equation. The results reveal that the dynamics experienced by the parcels constituting the updrafts near the SBF is almost the same as that ahead of the SBF, which explains why the strength of the updrafts near the SBF is similar to that ahead of the SBF.

In the typical convective boundary layer ahead of the SBF, the size and the strength of the updrafts scale with the boundary-layer height and the convective velocity scale, respectively, as is well known. Our results further reveal that this scaling also works for the updrafts near the SBF when the environmental wind is not included; however, this scaling breaks down when an environmental wind is included.

Surface heterogeneities can produce inland breezes (e.g., van Heerwaarden et al., 2014), which are also capable of triggering deep convection (Patton et al., 2005; Kang and Bryan, 2011; Rieck et al., 2014; Huang et al., 2019). Both sea breezes and inland breezes are produced by differential heating, so they are dynamically very similar. It is expected that the results in this study also apply to inland breezes.

## Code and data availability

The CM1 model is publicly available at https://www2.mmm.ucar.edu/people/bryan/cm1/getcode.html. Please contact S. Fu for the model output data.

## Author contribution

SF and RR designed the study. SF performed the simulations. All authors commented on the results and co-wrote the paper.

## Competing interests

The authors declare that they have no conflict of interest.

## Acknowledgements

We thank George Bryan for providing the CM1 model, and National Supercomputer Center in Guangzhou for technical support. This project is supported by National Natural Science Foundation of China (42105080 and 41930968). The National Center for Atmospheric Research is sponsored by the National Science Foundation.

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
