# Peer review of "Convective updrafts near sea-breeze fronts"

_Atmospheric Chemistry and Physics, 2022_

## Author Response (AR1)

We thank the editor and referees for the careful reading and constructive suggestions.

Below, the referees' comments are in **Black**, our responses are in **Blue**, and the changes in the manuscript are in **Red**.

**Response to referee 1:**

The manuscript examines the updraught strengths along and ahead the sea breeze front in an idealised LES setup with varying surface sensible heat flux over land and demonstrates why the updraughts are of similar strength but wider along the sea breeze front. It is a companion paper of another paper by Fu et al.

While the manuscript is well written and concise, the Figures are of good quality and it is reasonably convincing, it is still somewhat thin concerning parameter sampling and given that there is already a companion paper.

I found the discussion of older literature relatively thin, you can find more on it e.g. in Bechtold et al 1991, where also the effect on the background wind is discussed. In your results you simply dropped all discussion and experimentation on background wind and Coriolis scaling and I would ask at least for additional experimentation with varying background winds.

We do not expect the Coriolis force to have much effect on the updrafts due to the very small time and space scales of the convective updrafts. Fu et al. (2021) included the effect of Coriolis and found that the Coriolis effect on the updrafts is negligible.

We agree that the environmental wind exerts an important influence on the sea-breeze circulation, so we decided to perform some experiments regarding the environmental wind. Due to constraints of computational resources, we cannot explore a wide parameter space. Only two additional simulations, one with an offshore environmental wind of 2 m s$^{-1}$ and another with an onshore environmental wind of 2 m s$^{-1}$, are

conducted. Both simulations have a sensible heat flux of 0.2 K m s$^{-1}$. In these two simulations, the frontal updrafts are also larger than the prefrontal updrafts, and their dynamics is similar.

Please see Sect. 3.5 of the revised manuscript for the changes.

Also I found the discussion of boundary-layer rolls/streaks streaks l. 165-169 very thin, there is more literature on it including e.g on inflection point instability etc. Do these streaks or rolls depend on the surface fluxes over sea (set to zero in your experiment), more complicated in a moist problem, and the wind shear and will necessarily effect the updraughts. So you might also expand a bit more on this, possibly experimentally.

We added further discussions on the streaky structures, mentioning the frequently cited inflection-point instability and the convective instability (Etling and Brown, 1993). The inflection-point instability relies on the presence of inflection point in the $v$-wind profile. Figure 1 below shows the mean $v$-wind in a region behind the SBF, where streaky structures are visible. Such a weak $v$-wind is probably not able to support the formation of the streaky structures. The convective stability is usually measured with the parameter $-z_i/L$, where $z_i$ is the boundary-layer height and $L$ is the Obukhov length. As pointed out in the original manuscript, we found $-z_i/L$ is not able to predict the formation of streaky structures in our study. The parameter $-z_i/L$ is proposed in situations where the shear is limited near the surface; while in our study, the sea-breeze circulation produces a deep shear layer throughout the whole boundary layer, as shown in Fig. 2.

[Figure]

Figure 1. $v$-wind averaged from $x$ = 16 to 18 km and the whole $y$-direction at the end of the simulation in simulation SHF02. This region is behind the SBF, which is at $x$ = 20.1 km.

[Figure]

Figure 2. The same as Fig. 1, but for $u$-wind.

The sensible heat flux from the ocean is typically very small (Yu and Weller, 2007). For example, in the companion paper (Fu et al., 2021), the sensible heat flux over the ocean is found to be as small as 10 W m$^{-2}$. Therefore, we think the inclusion of ocean heat flux should not change the conclusions here. Rolls do form over the ocean, usually during cold-air outbreak events. These events are associated with strong offshore winds. In this situation, the sea-breeze circulation should not form.

References:

Etling, D. and Brown, R. A.: Roll vortices in the planetary boundary layer: A review, Boundary Layer Meteorol., 65, 215– 248, 1993.

Yu, L. and Weller, R. A.: Objectively Analyzed Air–Sea Heat Fluxes for the Global Ice-Free Oceans (1981–2005), Bull. Am. Meteorol. Soc., 88, 527–539, 2007.

The changes in the manuscript are at line 175 in the revised manuscript.

Previous studies have proposed several theories explaining the formation of streaky structures, e.g., the inflection-point instability and the convective instability (Etling and Brown, 1993; Gryschka and Raasch, 2005). The inflection-point instability relies on a sufficiently strong inflection point in the along-coast wind profile. However, no such inflection point exists in our simulations (not shown). The convective instability is usually measured with a parameter $-z_i/L$, where $L$ is the Obukhov length (Khanna and Brasseur, 1998; Salesky et al., 2017). In the region from $x$ = 10 to 12 km and from $y$ = -3 to 3 km, calculation shows that the mean value of $-z_i/L$ is 70. Based on previous studies, this value should correspond to cells, instead of the streaky structures shown in Fig. 3a. Previous studies suggest that the threshold values of $-z_i/L$ hold for situations where the shear is limited near the surface; while in our simulations, the shear occurs over a deep layer (Fig. 2c). It is interesting to mention that some studies suggest that wind shear alone is sufficient for the generation of streaky structures (e.g., Lee et al., 1990).

**Response to referee 2:**

This study is looking at updrafts around Sea Breeze fronts. I find the topic interesting, but much of the work here could do with a little bit more in-depth analysis. Some of the figures seem to have little added value over some of the others, and as a result the paper seems to mostly get stuck in a qualitative description of SBFs that has been known for a while. I would hope that the authors could sharpen the paper up a bit, with some suggestions below. Most importantly, I would be interested to see where a different updraft definition could lead, especially for a more robust statistical sampling. I would therefore recommend major revisions to this paper.

Methodological questions:

*) Lagrangian particles without a subgrid scale model. As suggested by L104, this would be not necessary for "large collections", but then the authors start tracing particles in a small single updraft

The two updrafts in Figs. 5a and 5b are sampled by 6260 and 3722 particles, respectively. As stated in the caption of Fig. 5, every 50th parcel trajectory is shown so that the individual parcel trajectories can be identified.

Based on Yang (2008), we expect the single-particle dispersion to be accurate for the purposes of identifying the main trajectories contributing to each updraft. In addition, Yang et al. (2015) also pointed out that the effect of subgrid-scale velocity on parcel trajectory should be small when the subgrid-scale TKE is much smaller than the resolved TKE, which is satisfied in our study. Based on these two references, the parcel trajectories should be reliable.

References:

Yang, F., Ovchinnikov, M., and Shaw, R. A.: Long-lifetime ice particles in mixed-phase stratiform clouds: Quasi-steady and recycled growth, J. Geophys. Res.-Atmos., 120, 11617–11635, https://doi.org/10.1002/2015JD023679, 2015.

The changes in the manuscript are at line 108 in the revised manuscript.

Yang et al. (2008) pointed out that the single-particle dispersion can be accurately modelled by LES because the errors in the Lagrangian velocity correlation and the Lagrangian velocity fluctuation tend to cancel each other. In addition, the model resolution in this study is so high that the subgrid-scale TKE is much smaller than the resolved TKE. In this situation, the effect of subgrid-scale velocity on parcel trajectory should be weak (Yang et al., 2015).

*) Only a small subset of the domain is initialized with particles, resulting in a depletion of the particle concentration and the need for a reset (together with the lack of subgrid model). Thus, particles end up being inhomogenously distributed across the updraft, and may cause biases.

Due to the fact that the updrafts are much narrower than the downdrafts in a convective boundary layer, it is probably unavoidable that the particles will be inhomogeneously distributed after about 10 minutes. We reset the positions of the particles to mitigate this effect, as mentioned in the original manuscript.

In order to reduce the computational cost and the need for storage space, we did not release particles in the whole domain. However, as shown below, the sub-domain where particles are released is large enough so that the updrafts being investigated are densely populated with particles by the end of the tracking. We note that the example below is representative.

[Figure]

Figure 3. (a) Horizontal and (b) vertical distribution of particles released at 30 min after release in simulation SHF02. The release time is at $t$ = 2 h 0 min and at $z$ = 0.51 km.

The changes in the manuscript are at line 118 in the revised manuscript.

This region is large enough so that the updrafts being investigated are always densely populated with parcels throughout the tracking.

*) The selection of updrafts purely based on their mid-BL w is a noisy way of doing it, with arbitrary tuning parameters, and assuming that a thermal extends through the boundary layer without tilt in this highly sheared environment (let alone a bubble vs plume discussion). What are the sensitivities to those parameters? And why not use buoyancy, or better yet an emitting/decaying scalar like Couvreux et al (2010 or so)?

We agree with the referee that some arbitrariness is introduced in our compositing technique, so we did some sensitivity tests regarding the parameters, i.e., the height and threshold vertical velocity that are used to identify updrafts. Note that in the original manuscript, we identify updrafts based on the vertical velocity at $z$ = $0.5z_i$ and the

threshold vertical velocity is $w = 0.8w^*$. Here, another two heights, i.e., $z = 0.3z_i$ and $0.7z_i$, and two threshold vertical velocity, i.e., $w = 1.0w^*$ and $1.2w^*$, are tested. It can be seen that the results are qualitatively the same.

[Figure]

Figure 4. (a) $x$-$z$ cross section and (b) $y$-$z$ cross section of the vertical velocity (m s$^{-1}$) of the composite frontal updraft in simulation SHF02. The updrafts are identified at $z = 0.3z_i$ and the threshold vertical velocity is $w = 0.8w^*$. (c) and (d) are the same as (a) and (b), except for the composite prefrontal updraft.

[Figure]

Figure 5. The same as Fig. 4, except that the updrafts are identified at $z = 0.5z_i$ and the threshold vertical velocity is $w = 0.8w^*$.

[Figure]

Figure 6. The same as Fig. 4, except that the updrafts are identified at $z = 0.7z_i$ and the threshold vertical velocity is $w = 0.8w^*$.

[Figure]

Figure 7. The same as Fig. 4, except that the updrafts are identified at $z = 0.5z_i$ and the threshold vertical velocity is $w = 1.0w^*$.

[Figure]

Figure 8. The same as Fig. 4, except that the updrafts are identified at $z = 0.5z_i$ and the threshold vertical velocity is $w = 1.2w^*$.

Our method does not assume that the updraft is always upright. As shown in Fig. 6a in the original manuscript, and the upper left panels in the sensitivity tests above, the composite updraft does tilt to the left.

We also agree that there are various ways of defining an updraft. In this study, the purpose is to understand the processes leading to deep convection. Previous studies showed that the initial lifting is critical to the initiation of deep convection. We believe that vertical velocity is the best variable that measures the initial lifting, so we define the updraft based on vertical velocity.

The changes in the manuscript at line 138 in the revised manuscript.

When defining the updrafts, we rely on two parameters, i.e., the height $z = 0.5z_i$, where the updrafts are defined, and the threshold vertical velocity $0.8w^*$, above which a grid point is defined as within an updraft. Sensitivity tests show that the results are qualitatively the same when the height is changed to $0.3z_i$ or $0.7z_i$, and when the threshold vertical velocity is changed to $1.0w^*$ or $1.2w^*$.

Content:

*) Fig 4: Not entirely sure what I am supposed to get out of this that isn't in Fig 3. Same for Fig 6: This seems to be the same information already in Fig 3?

Figures 3, 4 and 6 contain different, but complementary information. Figure 3 is a snapshot of horizontal and vertical wind at a single height $z = 0.5z_i$. It is used because it clearly shows the postfrontal streaky structures, and the spatial relation between the streaky structures and the large updrafts. However, we think it inappropriate to draw conclusions based on a single snapshot, so we used Fig. 4, where the information on all frontal updrafts is included. In addition, Fig. 4 is only a horizontal cross section at $z = 0.5z_i$. In order to get a complete picture of the updrafts, we also need to know the vertical cross sections, so we also used Fig. 6.

*) Fig 5: I'm not sure how the separation in 4 different groups happens exactly. Is this by 25%ile initial location along the x axis? Please describe this better. Also, the most important conclusion of Fig 5 seems obvious if only qualitative. (Near SBF parcels have the SB circulation superimposed on them). Is there a way to quantify this, in a statistical approach over many different plumes?

Yes, these parcels are divided into four groups by the 25th, 50th, and 75th percentiles of their initial $x$-positions, which is at $t = 2$ h 40 min.

We tried but failed to find a more quantitative way to present the results shown in Fig. 5.

Figures 3 and 4 have shown that the large updrafts are produced at the leading edge of the streaky structures, but do not show how the large updrafts are produced. So, we present Fig. 5 to show that the large updrafts near the SBF are produced by the merger between the updrafts originating from behind the SBF and the updrafts originating near the SBF. To the best of our knowledge, previous studies have not pointed out why the frontal updrafts are larger than the prefrontal updrafts.

The changes in the manuscript are at line 210 in the revised manuscript.

In order to know where the parcels come from, e.g., from behind the SBF or from ahead of the SBF, they are divided into four groups by the 25th, 50th, and 75th percentiles of their $x$-positions at $t$ = 2 h 40 min.

*) Fig 7: What is the added value of the Lagrangian approach here? The buoyancy and pressure gradient terms are also possible to calculate over a conditional average of the plume – with the bonus that it could cover all plumes in the entire Near/Far region. Or are the particles necessary to offset the challenges in the updraft definition above?

We think the Lagrangian approach is the better approach to analyze the dynamics of the updrafts. In principle, either the Lagrangian approach or the Eulerian approach is correct. However, if we simply composite the buoyancy and pressure gradient force, just like compositing the vertical velocity, we only know the information in the vertical direction but do not know the information in the horizontal direction. In this situation, we do not know the position of the particles, so we cannot know the vertical acceleration from the composite buoyancy and pressure gradient force. In other words, we do not know the "complete" dynamics experienced by the particles constituting the updraft. In the Lagrangian approach, the movement in the $x$-direction is implicitly included by the trajectory, so we know the vertical acceleration of the particle exactly.

*) Fig 9: If this is all dependent on very classical parameters, but not on the difference between those parameters (see for instance van Heerwaarden et al, JAS 2014)? Surely that should break at extreme values? Or is it because there simply was no ocean heat flux in this case?

The surface heterogeneity investigated by van Heerwaarden et al. (2014) is produced by variations in land cover or soil moisture. In that case, the warm patch and the cold patch can both have a non-negligible sensible heat flux. In our study, the heterogeneity is due to the land-sea contrast. It is known that the sensible heat flux from the ocean is usually very small (Yu and Weller, 2007). In the companion paper (Fu et al., 2021), the sensible heat flux over the ocean is found to be as small as 10 W m$^{-2}$. Therefore, we think it appropriate to neglect the sensible heat flux from the ocean surface.

As suggested by the other referee, we performed another two simulations regarding the environmental wind. It is found that the classical scaling no longer works when the environmental wind is included.

The maximum SHF in this study is 0.3 K m s$^{-1}$, which is close to the upper limit of SHF for most land surfaces. Therefore, the results in this study are relevant to the real atmosphere under most situations, even though the scaling might break down at much larger SHF.

References:

Yu, L. and Weller, R. A.: Objectively Analyzed Air–Sea Heat Fluxes for the Global Ice-Free Oceans (1981–2005), Bull. Am. Meteorol. Soc., 88, 527–539, 2007.

The changes in the manuscript are at line 92 in the revised manuscript.

Over the sea, the SHF is usually very small (Yu and Weller, 2007). Thus, a zero SHF is prescribed, as is done in previous idealized simulations of SBCs (Antonelli and Rotunno, 2007; Crosman and Horel, 2010, 2012).

Please also see Sect. 3.5 in the revised manuscript for the simulation results with environmental winds.